# Machine Learning of Raman Spectroscopy Data for Classifying Cancers: A Review of the Recent Literature

**DOI:** 10.3390/diagnostics12061491

**Published:** 2022-06-17

**Authors:** Nathan Blake, Riana Gaifulina, Lewis D. Griffin, Ian M. Bell, Geraint M. H. Thomas

**Affiliations:** 1Department of Cell and Developmental Biology, University College London, London WC1E 6BT, UK; nathan.blake.15@ucl.ac.uk (N.B.); r.gaifulina@ucl.ac.uk (R.G.); 2Department of Computer Science, University College London, London WC1E 6BT, UK; l.griffin@ucl.ac.uk; 3Spectroscopy Products Division, Renishaw plc, Wotton-under-Edge GL12 8JR, UK; ian.bell@renishaw.com

**Keywords:** Raman Spectroscopy, medical application, disease screening and diagnosis, machine learning, cross-validation, deep learning

## Abstract

Raman Spectroscopy has long been anticipated to augment clinical decision making, such as classifying oncological samples. Unfortunately, the complexity of Raman data has thus far inhibited their routine use in clinical settings. Traditional machine learning models have been used to help exploit this information, but recent advances in deep learning have the potential to improve the field. However, there are a number of potential pitfalls with both traditional and deep learning models. We conduct a literature review to ascertain the recent machine learning methods used to classify cancers using Raman spectral data. We find that while deep learning models are popular, and ostensibly outperform traditional learning models, there are many methodological considerations which may be leading to an over-estimation of performance; primarily, small sample sizes which compound sub-optimal choices regarding sampling and validation strategies. Amongst several recommendations is a call to collate large benchmark Raman datasets, similar to those that have helped transform digital pathology, which researchers can use to develop and refine deep learning models.

## 1. Introduction

Biomedical applications of Raman Spectroscopy (RS) have been steadily growing over the decades as technology matures. Applications range from the label-free staining of histology slides to determine the biochemical composition of a sample to classification tasks to determine the presence and grade of disease, including determining tumour margins during cancer surgery [1]. RS has the potential to serve two clinical needs in the domain of oncology. The current standard for cancer diagnosis is the assessment of excised suspicious tissues by a histopathologist. Despite the high level of training of such professionals, this assessment contains a degree of subjectivity leading to inter- and intra-observer variability [2,3,4]. It is hoped that RS can provide an adjunct to the histopathologist to reduce this variability. Additionally, most cancers develop through pre-malignant stages and the treatment of these early pathologies can prevent development into malignant cases [5]. Such early changes are often subtle and it is hoped that RS may enhance our ability to detect pre-malignant and early-stage cancers. Both of these needs can be expressed in terms of a classification task.

Machine learning (ML) has long been used to classify Raman spectral data. The key feature of ML is that a model in some way learns from data. It can therefore be described as a data-driven approach to modelling. Although there are many variations, the most common ML model used in biomedical RS is Principal Component Analysis–Linear Discriminant Analysis (PCA-LDA) [6]. PCA reduces the dimensionality of the data and removes some noise; LDA then learns a criterion by which to separate data as belonging to one of several classes, based on labelled examples. This is one example of a traditional ML model, of which there are many. These are held in contrast to deep learning models, which are large and complex models based on a neural network architecture. In particular, we define a deep learning model as any model based on a neural network architecture, from artificial neural networks (ANNs) to more sophisticated structures such as convolutional neural networks (CNNs). Traditional ML models include all other models, whether they are linear, such as LDA, or non-linear, such as support-vector machines (SVMs) with an appropriate kernel function. Deep models could revolutionise the digital healthcare space [7], including biophotonics [8]. In particular, their ability to capture non-linear complexities in a dataset allows them to exploit patterns too subtle for traditional methods, making them an ideal candidate to realise the full potential of RS. An area that has already benefited from deep learning is digital pathology, which is often applied to oncology [9].

However, ML, traditional or otherwise, is not without its limitations. Just as medical researchers need to understand something of the statistical science of hypothesis testing, and the debate and misunderstandings regarding *p*-values, it is becoming increasingly important to become literate in ML [10]. One of the barriers to transferring promising ML results to clinical settings is the reproducibility of results [1]. Indeed, a recent review of ML applications to diagnose COVID-19 using chest radiographs or CT scans found that of sixty-two studies, none were of sufficient quality to be clinically relevant [11]. Prominent among the given reasons were methodological issues that compromise the generalisability of a model to the target population. Aside from modelling issues, a number of practical issues remain, such as establishing cost-effectiveness, the choice of substrate on which to mount ex vivo samples and the miniaturisation of the technology for in vivo testing while maintaining the signal-to-noise ratio [12]. Though important for RS to become established in clinical practice, these will not be discussed in this review.

### Assessing Model Performance

The pertinent point of interest in how well an ML model performs is how it would cope with previously unseen data in the clinical setting: its generalisability. It is possible for a model to simply memorise data, thus giving a perfect classification for data it has seen with no ability to generalise to unseen data. This is known as over-fitting. Ideally, performance would be tested with a newly created dataset. However, there are many practical limitations to collecting new data, particularly in clinical research, which can be expensive and time-consuming. A common compromise is to split the existing dataset into a training and a test set; the former being used to create the model, the latter simulating the process of collecting a new dataset and being used to test the models performance. This requires holding out a proportion of the data during training. This can compound the problem of small datasets. Therefore, an extension of single train/test splitting is *k*-fold cross-validation (CV). This repeats the training/test split *k* times such that all of the data are sequentially used in the test set, thus producing *k* estimates of the models performance. The average performance is then given, sometimes with an accompanying measure of variance. Taken to the extreme is leave-one-out CV (LOOCV), in which the test set comprises a single data point, thus training the model on the maximum possible amount of data.

In addition to the training/test split, sometimes an additional split is made called the validation set. The validation set is used to optimise the model hyper-parameters (or even guide the choice of ML model): choices about the model that a researcher makes which influence its classification ability. Similar to how a model can over-fit the data, the hyper-parameters can be selected such that it performs well for a given test set, but fails to generalise well. This has been described as over-fitting at the second level of inference [13]. The validation set provides an additional set to allow the hyper-parameter space to be optimised, while preserving the test set for a less biased estimate of the generalisability.

Regular CV splits the data such that the same test data are never used twice. Repeated CV iterates this process several times such that many permutations of possible test sets are used (the test set being comprised of multiple samples, except for in LOOCV). It is not commonly used as it is computationally expensive, though it can reduce the variance of a model’s estimated performance when sample sizes are small [14].

## 2. Materials and Methods

### 2.1. Literature Search

This literature review follows the principles set out in the PRISMA (Preferred Reporting Items for Systematic reviews and Meta-analyses) guidelines [15]. The databases PubMed and Web of Science were extensively searched by combining the search terms ‘Raman Spectroscopy’ and ‘Learning’ with the AND Boolean operator. Titles and abstracts in the databases were thus searched, as illustrated in Figure 1, and oncology studies were identified. Publications were limited to the English language and being published from January 2018 to the date of the search (October 2021). Potentially relevant studies were selected for a full text review. Additional studies were identified among the references of identified studies. Studies were excluded if they did not explicitly classify data or were not peer reviewed. As the ML methodology was itself the focus of this review, no attempt to exclude studies based on methodological quality was made and so the PRISMA quality checklist was not applied. Studies involving surface-enhanced Raman Spectroscopy were excluded.

Many studies used several ML models, conducted analyses on different subsets of their data and/or compared several pre-processing techniques, producing a multitude of results. For instance, several studies compared the performance of different machine learning models, often traditional ML models such as LDA against deep learning ML models, such as CNNs. For ease of comparison, and to mitigate selection bias on our part, the best-performing model and/or dataset is presented in Table 1; other results are included when pertinent to a particular discussion. In the vast majority of cases, the accuracy of a model was the primary reported performance metric: the number of correct classifications divided by the total number of classification attempts. Although its suitability to prediction tasks has been questioned, because of its ubiquity in the reviewed literature and its intuitive interpretation, we report this metric unless otherwise stated.

### 2.2. Data Collection

From each study, we extracted: (1) authors, institution and year of publication; (2) type of cancer and sample substrate; (3); ML models used; (4) validation strategy; (5) number of patients and samples; (6) number of spectra; (7) how the data were split during validation; (8) number of classes classified; (9) performance metrics.

## 3. Results

A total of 25 studies were identified (Table 1), 18 of which interrogated tissues, 5 blood serum and 2 studied cell lines. All of these studies classified Raman spectra into at least two groups, usually healthy and cancerous. The cancers explored in the literature included brain (5), tongue (3), breast (3), skin (3), lung (2), prostate (2), nasopharyngeal (2), colon (2), oral (1), cervical (1), ovarian (1) and kidney (1).

### 3.1. Oral and Nasopharangeal Cancers

Xia et al. [36] probed tongue squamous cell tissues using a fibre optic Raman spectrometer and developed a CNN-SVM for binary classification. This model replaces the final dense layer of a typical CNN with an SVM, combining the feature selection prowess of the former with the classification abilities of the latter. SVMs can utilise a number of kernel functions to better model non-linearities in the data; in this paper, a radial basis function (RBF) was used. They compared this model to a standard CNN as well as PCA-LDA and PCA-SVM(RBF). The CNN-SVM performed best (accuracy = 99.54%; sensitivity = 99.54%; specificity = 99.54%), as determined by accuracy, though trade-offs between sensitivity and specificity may change this interpretation according to clinical needs.

The same team used a similar set-up to collect two datasets taken under conditions of ‘illumination’ and ‘no light’ [37]. These datasets underwent a further division of pre-processing or no pre-processing, to make for four datasets. These were used to classify spectra into binary classes using an ensemble CNN, in which several CNN models were trained and the outputs integrated to give a consensus. They found that the best performance was attained under the no-ambient-light conditions with pre-processing (accuracy = 98.75%; sensitivity = 99.10%; specificity = 98.29%), although the difference in accuracy to the worst-performing dataset (illumination and no pre-processing) was only 4.75%.

The last publication from this team used a similar set-up and dataset to compare the performance of a custom-built CNN against PCA-LDA and PCA-SVM (with a radial basis and a polynomial kernel) [38]. They found that the CNN outperformed the other ML models (accuracy = 96.90%; sensitivity = 91.67%; specificity = 94.44%).

It is not clear if the tissues used in these three studies are the same. However, in all cases the diseased and healthy samples were obtained from the same subjects.

Also investigating oral cancers, Jeng et al. interrogated cryopreserved samples, seeking to discriminate between healthy and cancerous tissues [24]. They further performed a sub-group analysis, dividing their dataset into tongue, buccal and gingiva tissues to perform three pairwise cancerous versus healthy binary classifications. They additionally performed a ‘point-wise’ approach in which five spectra were taken per sample and a ‘patient-wise’ approach in which the average of these five spectra was taken. They explored two CV techniques, comparing a *k*-fold versus a LOOCV strategy. Finally, they compared a PCA-LDA and a PCA-QDA (Quadratic Discriminant Analysis) classifier. Using these methods, they found that taking the average spectrum of a sample yielded a better performance than a point-wise approach and with PCA-QDA typically performing better than PCA-LDA, though not across all sub-group analyses. LOOCV resulted in lower error rates compared to *k*-fold CV for the ‘all cancer’ versus ‘healthy’ analysis, but this was reversed for the sub-group analysis, which consisted of smaller sample sizes.

Two studies focused on nasopharyngeal cancers. Zuvela et al. used an in vivo set-up to collect data during endoscopy [40]. They employed a genetic algorithm (GA) to perform feature selection for a PLS (Partial Least Squares)-LDA binary classifier, comparing its performance to a PLS-LDA model without this selection. They also compared performance when utilising either the fingerprint region or the high wavenumber region, or both combined. Not only did the GA-PLS-LDA outperform the generic model (accuracy = 98.23% versus 95.58%), but this feature selection was also used to find the candidate Raman peaks responsible for this discrimination. Additionally, though combining fingerprint and high wavenumber regions may actually confuse ML models by including irrelevant data, the GA feature selection was able to mitigate against this potential pitfall, improving accuracy (fingerprint region = 92.04%; high wavenumber = 94.69%; both = 98.23%).

The same team expanded their investigations with a similar study which included many more subjects, samples and spectra with a similar recruitment protocol [34]. Of all the studies reviewed, this is the largest in terms of the number of subjects recruited. For this study, they used a CNN to classify three classes (cancerous, post-treatment and healthy) in a pairwise fashion. This consistently performed better than a PLS-LDA classifier. The CNNs’ superior performance was maintained even when the sample size was down-sampled by factors of two and four, contrary to the idea that CNNs require an abundance of data from which to learn. Indeed, the CNN trained on data down-sampled by a factor of two produced the best performance.

### 3.2. Lung Cancers

Qi et al. adopted a novel approach to classify Raman spectra of lung tissue as adenocarcinoma, squamous cell carcinoma or normal in a pairwise fashion [29]. They transformed the data into 2D spectrograms in a similar process used to classify audio data. These spectrogram data was used in a CNN accepting 2D inputs, akin to typical image classifiers and different from all the 1D inputs thus far discussed, and compared performance to a PCA-LDA model. For both pairwise comparisons the CNN returned an accuracy over 96%, while neither PCA-LDA model broached 90%.

Chen et al. also discriminated between lung cancers, as well as glioma, a common brain cancer, using spectra taken from blood serum [20]. They compared both classes against healthy controls in a pairwise manner. Several deep learning architectures were compared: an ANN, a RNN (Recurrent Neural Network), a LSTM (Long Short-Term Memory) and AlexNet (a particular architecture of CNN). The dimension reduction techniques of PCA and PLS were also compared to no pre-treatment. The use of data augmentation was also explored by increasing the number of spectra 5-fold. The augmentation details are discussed under the ‘Data Augmentation’ section. Across all analyses, this augmentation increased performance. This was most pronounced when PLS was first performed on the data. AlexNet, the largest model used, together with PLS and data augmentation, was the best-performing model, although the difference amongst all the models, except the ANN, was minimal. However, when a three-class model was constructed, the best performance had an accuracy of 85.1%.

### 3.3. Brain Cancers

Two studies from the same team also explored gliomas. Riva et al. took fresh-tissue biopsies and classified healthy versus cancerous tissue using the traditional models of Random Forest (RF) and Gradient Boost Tree (GB) [30]. The latter model performed best because its feature selection allowed the detection of novel Raman peaks to be implicated in gliomas. In the team’s second study, Sciotino et al. explored the potential to discriminate between the mutational status of gliomas, essentially attempting to genotype using RS [32]. They used GB and SVM (RBF) to successfully classify between the two disease genotypes.

Bury et al. also analysed brain tissue, attempting to discriminate the primary source of metastatic brain cancers [18]. Seven samples each with primary sources of lung adenocarcinoma, colorectal carcinoma and melanomas were obtained and 25 spectra collected per section. RS was compared to attenuated total reflection-Fourier transform infrared (ATR-FTIR) spectroscopy. An overall accuracy of 69.7% was achieved compared to just 55.3% using similar PCA-LDA modelling on ATR-FTIR data. These improved when the two adenocarcinoma categories were merged into a single group to 80.2% and 84.0%, respectively.

Mehta et al. used 35 serum samples from meningioma patients and compared them to 35 samples from healthy controls in an attempt to develop an approach to diagnose brain tumours using minimally invasive techniques [28]. Approximately eight spectra were taken per sample, and the average of these was used for analysis. Employing PCA-LDA they achieved an accuracy of 86% for discriminating meningioma from healthy samples, which fell to 70% when the model was tested against an independent held out test set.

### 3.4. Breast Cancers

Koya et al. created Raman maps from ex vivo breast tissue and classified spectra as cancerous or healthy [25]. It is the largest study in terms of Raman spectra, though not samples; the Raman mapping methodology allowed them to take many spectra per sample. A CNN was used to classify the spectra. They used a technique called ‘permutation importance’ to interpret the CNN outputs and find which Raman bands were biologically significant.

Ma et al. also classified breast tissue using a CNN [27]. They compared its performance against four SVMs (each with a different kernel) and Fisher’s Discriminant Analysis (FDA). Data augmentation was required to improve the CNN from the worst to the best-performing model.

Zhang et al. interrogated five breast cancer and one healthy breast cell lines with RS [39], using PCA-DFA (Discriminant Factor Analysis) and PCA-SVM to classify spectra. The latter technique in particular was well able to separate healthy from cancerous cell lines with an accuracy of 99.0%. The team also performed a number of clinically relevant sub-group analyses and still achieved an accuracy of 93.9% with a four-class model. Performance deteriorated as the sub-class divisions became more nuanced, representing comparisons between ever more biochemically homogenous samples.

### 3.5. Prostate Cancers

Lee et al. explored Raman spectra of extracellular vesicles derived from blood serum samples as a biomarker for prostate cancer in combination with a CNN [26]. This was compared to PCA-LDA and PCA-QDA models. Additionally, analyses were performed on three wavenumber regions (full spectrum, fingerprint and high wavenumber regions), and with the data in its raw form as well as pre-processed. The CNN outperformed the traditional ML models across all subsets. The fingerprint region generally leant to better performance, though that was not ubiquitous across all subset analyses.

### 3.6. Gastrointestinal Cancers

Wu et al. interrogated biopsy samples taken during endoscopy, classifying spectra as normal, adenomatous polyps or adenocarcinomas [35]. They found that a CNN comprehensively outperformed several traditional ML models. They also explored the difference when conducting analysis on pre-processed versus just normalisation data. Finally, the team performed CV via two methods, one splitting at the level of spectra, the other splitting at the level of subject/sample (there was one sample per subject so these coincide). The former method achieved an accuracy of 93.8%, falling to 81.3% with the latter split.

Ito et al. developed a boosted tree model from serum samples taken from suspected colorectal cancer patients, classifying them into four categories, colorectal cancer, adenoma, hyper-plastic polyps and neuro-endocrine tumours, in a pairwise fashion [23]. They achieved 100% accuracy in all tasks, although they used the $R^{2}$ value as their assessment metric which gives a more nuanced idea of performance by accounting for how certain the model was in its classification, punishing predictions further from the class label. By this metric, the boosted trees still performed exceptionally well. It is, however, unclear whether there was any validation/test set, and so these results may reflect the training performance.

### 3.7. Skin Cancers

Serzhantov et al. used an ensemble of traditional ML models to classify skin tissue as cancerous or normal [33]. The models included were a classification and regression tree, SVM, *k*-nearest neighbours and logistic regression. Instead of selecting the best single model, the outputs of all models were used to create a soft voting classifier, allowing each model to ‘vote’ on an outcome, and the consensus across all models was taken. Splitting the data 50/50 into train/test sets, this was repeated 1000 times to build a spread of estimates. This method achieved an accuracy of 90.5%.

Baria et al. compared PCA-LDA and PCA-ANN for the task of classifying spectra taken from cultured cell lines to distinguish between three skin melanoma genotypes [17]. The LDA produced an accuracy of 92.7%, and the ANN 96.7%.

Santos et al. classified skin samples with spectra from the high wavenumber region using PCA-LDA, distinguishing between melanoma and not-melanoma [31]. They achieved an accuracy of 62.5%. The classification model was used in a unique way: they took the LDA score outputs and, instead of setting a typical limit of 0.5 as the delineation score between melanoma or not, chose a criteria of any two spectra from a single sample having a score greater than 0.35, or any single spectrum having a score greater than 0.8.

### 3.8. Gynaecological Cancers

Daniel et al. compared a PCA-LDA model to a PCA-ANN model in classifying cervical tissue as healthy, neoplastic or malignant [21]. In addition, those samples determined to be malignant were then subject to another LDA model to determine whether the samples were well, moderately or poorly differentiated. The PCA-LDA model achieved an accuracy of 95.3% compared to 99.0% for the PCA-ANN model. To help determine the biochemistry that characterised the three classes, non-negative least squares (NNLS) was used to fit eleven known biochemical signatures to the spectra. This provides a multivariate method of determining sample biochemistry compared to the usual univariate peak assignment method.

Chen et al. used RS on serum to classify ovarian samples as normal, cystic or cancerous in a two-step binary classification regime [19]. The first step used an ANN to determine abnormal and healthy samples, using an ensemble method to select the best model architecture, with an accuracy of 94.8%. Abnormal samples were then entered into another ANN to determine whether they were cystic or cancerous, achieving an overall accuracy across the three classes of 86.2%.

### 3.9. Other Cancers

He et al. interrogated ex vivo renal tissue seeking to identify cancerous tissue and demarc surgical boundaries as well as classify those tissues [22]. Although 100 spectra were obtained per sample, only 30 were used for classification after saturated spectra were removed. They used a suite of ML models, with an SVM (RBF) model marginally outperforming an ANN, while distinguishing between cancerous, normal and fat tissues with 92.89% accuracy, with only a slightly lower performance when classifying cancerous sub-types.

## 4. Discussion

### 4.1. Validation Strategies

Nearly all of the reviewed studies conducted some kind of splitting of the data in order to produce train, test and, in some cases, validation sets. Several partitioning strategies were used (Figure 2). The most common were LOOCV and *k*-fold CV, which were either 5-fold or 10-fold. These are common default values in ML as they have been found to produce a good balance between bias, variance and computational cost [14]. However, they are somewhat arbitrary choices and what would constitute the optimal strategy is a nuanced topic. LOOCV theoretically should provide the least biased performance, as it incorporates the largest amount of training data, with the lowest variance, as it only shifts one sample across folds [41]. However, this assumes that CV is averaging independent estimates, but the samples may in fact be highly correlated. LOOCV could then struggle to detect model instabilities caused by changing the dataset, as only one sample is changing at a time. Consequently, a *k*-fold CV strategy may be preferable, though the precise number of *k* depends on a number of interacting factors such as the sample size, signal-to-noise ratio of the data and the model used, which are not easy to reconcile. Most of the reviewed studies had low sample sizes, with an average of 82 subjects. When the sample size is small, LOOCV has been shown to have a high bias and variance, while *k*-fold strategies had a low bias and their variance can be further reduced by performing repeated splits [42]. One of the reviewed studies did explore the difference in performance based on a LOOCV versus *k*-fold CV strategy. Jeng et al. found that LOOCV yielded higher accuracies than *k*-fold CV (the value of *k* was not specified) for a binary classification task, but this was reversed in a three-class problem. The study did not compare the variance or bias of these performances.

Standard *k*-fold CV splits data into folds at random. Shu et al. employed a variant called Venetian Blinds, which systematically assigns data to folds [34]. There is some suggestion that the method performs well compared to random and other methods, though this depends upon the ML model used [43], and very likely the dataset in question. Overall, it has received little attention in the literature, and its utility to Raman datasets unexplored.

Repeating CV several times has been shown to be effective at reducing the variance of generalisation estimates [14]. Serzhantov et al. repeated a 50/50 training/test split 1000 times [33]. Unfortunately, their purpose was not to investigate the effect repeated CV had on the bias or variance of the generalisation error and so was not explored. This could have been estimated by comparing models trained on partitions of the data to a model trained on an entire dataset [14].

Single-split validation refers to the method of dividing the data into different sets just once, as opposed to CV methods which iteratively split the data to give several performance estimates. Although single splits have been shown to be unbiased, they have a high variance, particularly with small datasets [42]. Single splits are often unpopular in medical ML applications with small sample sizes due to the technique not utilising a certain proportion of the data for training. Here, seven of the reviewed studies used single data splits, suggesting it remains a common strategy in biomedical RS applications.

#### 4.1.1. Partitioning Data with a Hierarchical Structure

Most of the reviewed studies classified individual spectra, while some classified average spectra [23,28]. Many such spectra were often taken from the same sample, and several samples were sometimes taken from the same subject. This introduces a hierarchical structure to the data with subjects at the top, followed by samples and then the spectra themselves. This introduces a complexity to medical Raman datasets which needs to be taken into account. For instance, it raises the question of which level to split the data: spectra, sample or subject (Figure 3). If split at the level of spectra, this could mean that spectra belonging to the same sample and/or subject are present in both the training and test set. This could lead to overly optimistic estimates of the generalisability of the model as it is not a realistic assessment of the model, which would be classifying spectra from unseen subjects in the clinical setting.

Of those studies that split the data at the level of spectra, the best accuracies of 90%, 96.6%, 97.7%, 93.8% and 94.8% were achieved [19,25,26,29,35]. Of those studies splitting data at the level of subject or sample, the accuracies were: 84.4%, 99%, 81.8%, 98.3%, 83%, 87%, 92.9%, 81.3% and 62.5% [20,22,24,30,31,32,34,35,40]. Of those studies in which the level of split was not explicitly stated, the accuracies were: 86.0%, 96.7%, 80.2%, 99.0%, 92.0%, 90.5%, 99.5%, 98.8%, 96.9% and 99.0% [16,17,18,21,27,33,36,37,38,39]. No attempt has been made to statistically compare these groups, as might be performed during a meta-analysis, as the various study aims and methodologies are too heterogeneous to make such comparisons statistically valid. However, it can qualitatively be seen that studies which split at the level of subject or sample tend to report lower accuracies than those split at the level of the spectra, or do not explicitly state the level of the split. This likely reflects more realistic assessments of how well the model would perform in the clinical setting. Of particular note is the study by Wu et al., the only reviewed study which compared methods of splitting the same dataset. They found a drop in performance of 12.5% in the overall accuracy when splitting at the sample level compared to the spectra level [35]. This is consistent with the findings from Guo et al., who explicitly examined the difference the level of the split makes during CV with tumour cell lines, concluding that the highest hierarchical level of the dataset should be used when partitioning the data [44].

For many studies, one sample was taken per subject. However, some studies took multiple samples from the same subjects, which introduces an additional strata into the the hierarchy. For instance, Zuvela et al. took 113 samples from 60 patients [40] and Shu et al. sampled 888 sites from 418 subjects [34]. Both studies split the data at the level of samples rather than the highest level, subject, so it is not possible to ascertain what impact this may have had on subsequent analyses.

Two studies took only the average spectrum from a single sample, thus flattening the hierarchical structure of the data and bypassing this issue [23,28]. Both studies took spectra from serum samples and analysed the data using traditional ML models. Neither study examined how taking the average spectra per sample compared to using all sample spectra. Jeng et al. compared the performance of using average spectra versus all five spectra of a single sample in their PCA-QDA model, finding that the former method had an accuracy of 88.75% versus 83.00% [24].

There are two methods by which sample heterogeneity is currently incorporated in the literature. Averaging spectra provides one means, while including multiple spectra from a single sample provides another. There has been no direct comparison of these methods in the reviewed literature. A priori, there is no reason to suspect one will work better than the other and will likely depend upon the intended application. For instance, a mean spectrum typically has a higher signal-to-noise ratio than individual spectra, and a model trained on the former may not generalise to applications requiring individual spectra to be classified, such as post-surgery cancer edge detection. It is also an open question whether averaging spectra from a sample before analysis would be an effective use of data for deep learning models, which are notoriously data intensive. Overall, it is not clear that averaging spectra will always provide a benefit, and the decision to do so should take into account the nature of the application and the ML model being used.

#### 4.1.2. Paired Sampling

Many studies took healthy and diseased samples from the same patient [24,25,27,30,32,36,37,38]. This is completely understandable given the ethical constraints upon taking healthy tissue, particularly with neurological tissues, but the consequences this has upon generalising to unseen patients needs exploring. Ma et al. argue that this ‘paired’ sampling reduces ‘interference caused by individual differences’ [27]. This may be true of traditional statistical set-ups, such as hypothesis testing, but does not necessarily extend to ML. In hypothesis testing, paired sampling allows a single mean and variance to be calculated: that of the difference between paired samples. A similar procedure is not typically conducted in ML, including the literature reviewed here. Rather, even if paired samples have been taken, they are treated as independent samples (analogous to paired samples being taken but a normal, non-paired, statistical hypothesis test being performed). Additionally, by using paired data, it could be argued that the training sample is not as inclusive of the general population, denying the model the opportunity to distinguish ever more subtle differences between cancerous versus healthy tissues. Again, this depends upon the clinical application: will classification be made on the basis of comparing a patient’s healthy sample to their own suspected diseased sample? If not, then a paired training regime does not reflect the clinical application, and will compromise generalisability. However, ML models could be adjusted to account for paired samples, and may be particularly useful for longitudinal studies and disease progression tracking, where the task is explicitly to compare a single patient’s sample against their past samples.

#### 4.1.3. Sample Representativeness and Label Noise

Some studies were able to obtain healthy spectra from in vivo tissue, usually via endoscopy [34,35,40]. The determination of healthy tissue is made by an expert operator (i.e., an endoscopist) at the time of the examination, while suspicious tissues are excised and sent for a more thorough histopathological examination. This means that diseased tissues have been more thoroughly examined than their healthy counterparts. This is compounded in RS studies as the technology is able to detect pathological biochemical changes that have yet to manifest in the tissue morphology [45]; hence, tissue determined as healthy based on morphology alone may be contaminated with pre-clinical pathological signals. In the context of ML, this is understood as label noise which describes not noise in the data themselves but in the labelling of these data [46]. This potential mislabelling of healthy samples can be described as ‘noisy at random’, in which the source of noise is dependent upon the true classes. Due to practical and ethical constraints, some degree of contamination is inevitable, and this should be factored in when considering the implications for any results. The primary concern in this context is considering how representative the data are of the population in which they will be deployed.

Wu et al. bypassed this potential problem by obtaining independent samples by including patients who had biopsies taken of suspicious lesions detected during endoscopy, which were later determined to be normal by traditional histopathology [35].

There are other potential sources of label noise. It could occur when Raman maps are taken of an area which are given a single label, as Koya et al. did, but it is possible that the area is not homogenous in its class, thus mislabelling some spectra. This is only a problem if spectra, rather than the entire map, are being classified, and might be mitigated if average spectra are taken, to the degree that one class dominates the sample. Santos et al., as well as taking the average sample spectra for training, also explicitly controlled for this by designating certain samples as heterogeneous during histopathological assessment [31]: samples deemed to have an uneven distribution of histological features. To control for this, during train/test splitting, they ensured that both homogenous and heterogenous samples were included in both sets. They then used the homogenous training set to build a PCA-LDA model and used the heterogenous training set to define the parameters of the diagnostic model (i.e., to interpret the outputs of the model). This model achieved a sensitivity and specificity of 100% and 43.8%, respectively. This is comparable to a 2016 study by the same team on a similar dataset but using only the homogenous samples, which achieved a sensitivity and specificity of 100% and 45% [47].

Another source of label noise that is perhaps more well understood comes from mislabelling amongst domain experts such as histopathologists. The negative impact of inter-observer variation upon ML models has been demonstrated in the context of circulating cancer cells [48]. A way to improve label accuracy is to obtain consensus pathology, where more than one pathologist classifies the same samples, as did some of the reviewed studies [31,34,40]. This may entail discarding valuable medical data if a consensus cannot be reached. Additionally, how consensus pathology is utilised needs careful consideration. If only data achieving a consensus label are included to train a model, this has the effect of enriching the dataset with less ambiguous class examples. This would limit the generalisability of the model when deployed in the clinical setting where it would undoubtedly encounter more ambiguous samples. This reflects the fact that disagreement amongst expert medical annotators is clinically relevant; the boundaries between classes, for instance, high- versus low-grade dysplasia, exist on a continuum. This ambiguity in the data could be incorporated into a model by using fuzzy classification, where class membership is not binary. Other data-retaining options include utilising ML models known to be robust against class noise, re-weighting data towards clean labels, and more sophisticated methods specific to deep learning such as teacher and student models [49].

### 4.2. Data Augmentation

Data augmentation is a technique by which the amount of spectra is increased by adding replicated spectra to the data and adding noise and/or other alterations to them. This is a technique used in deep learning to both increase the sample size and to inject noise into the data so that the model is less likely to overfit to the training set. Not only does this increase the sample size, which is important for data-intensive deep models, but it also makes the model robust against irrelevant features in the data by forcing it to pay attention to what is not being distorted. This has the effect of regularising the data, smoothing out the learning process. This is well established and widely used in other fields using deep learning, particularly image recognition [50]. This process does not seek to simulate biological variation, but is useful as a regularisation technique, which lessens the tendency of a model to over-fit the data. Hence, augmentation cannot be used to compensate for a lack of biomedical diversity in the data.

Of the studies reviewed here using deep learning, six employed data augmentation. This was achieved via several strategies. Chen et al. added white Gaussian noise of varying levels to the spectra, increasing the training data by a factor of five [20]. Lee et al. similarly added Gaussian noise to spectra, increasing the entire dataset by a factor of four [26]. Ma et al. also added random Gaussian noise in addition to shifting the wavenumber axis up to 2 cm${}^{-1}$ and adding a random scale coefficient, thus increasing the sample size from 600 to 5000 spectra [27], increasing accuracy from 75% to 92%. Wu et al. also performed wavenumber shifting, up to 4 cm${}^{-1}$, as well as adding linear combinations of 2–5 random spectra from the same class to create a new spectrum, thus increasing the sample from 233 to 2420 spectra. Fang et al. linearly combined several spectra to create a new spectrum and also performed ‘wavenumber shifting’ and added ‘random noise’, creating 6600 spectra from 510 spectra [51]. Xia et al. augmented the training set up to an unspecified number, shifting the wavenumber axis and adding noise to the magnitude at each wavenumber, a process which more closely resembles the Poisson noise typical of Raman spectra, compared to adding Gaussian noise [36]. Shu et al. developed a novel augmentation strategy, flipping spectra both vertically and horizontally as might be done in typical image data augmentation [34].

Only two studies assessed the impact data augmentation had upon classification performance. Chen et al. found it consistently increased performance across multiple subset analyses [20], and Ma et al. found it increased the overall accuracy from 75% to 92% [27].

Augmentation is traditionally only performed upon the training data, as data inflation and its regularisation effect is only pertinent during training. The test data are a representation of the general population of interest and adding noise could be considered dangerous as it may then become less representative. Four of the reviewed studies which performed data augmentation did so on the entire dataset before splitting into training and test sets [26,27,35,51]. There is a technique called test-time augmentation which is becoming more common, particularly with small datasets. It has been shown to increase model performance [52]. It allows multiple predictions on the same test spectrum, augmented several times, of which an average can be taken—essentially creating an ensemble approach. Four of the above studies applied data augmentation to the entire dataset, conducting de facto test-time augmentation. However, it is not clear from their methods that they exploited the merging of predictions.

### 4.3. Pre-Processing

There are numerous pre-processing techniques involved in the analysis of Raman spectra, such as baseline correction, smoothing and normalisation. However, one of the putative benefits of CNNs is that they can automatically perform feature selection and pre-processing at the same time. There is a degree of arbitrariness to many pre-processing steps, with evidence suggesting that most pre-processing techniques, and their numerous parameters, actually worsen the subsequent classification [53]. Finding the best pre-processing method and parameters is often a case of trial and error, or relying on what has worked well in the past. Although more systematic approaches exist, such as searching with a genetic algorithm [53], removing this step is attractive. However, neural networks in general require normalisation to avoid problems such as exploding or vanishing weights; hence, in the subsequent discussion references to ‘raw’ data includes normalisation. Three studies explored a suite of pre-processing steps against raw data.

Lee et al. compared ‘baseline corrected’ data to raw data and found that for traditional ML models baseline correction improved performance [26]. However, the CNN performed better on the raw data with an accuracy of 96.6% +/− 0.9% compared to 90.2% +/− 0.5%. The best performance on pre-processed data was PCA-QDA with an accuracy of 95.0%.

This suggests that CNNs can classify data without pre-processing. There is even a suggestion that the architecture is able to exploit diagnostic information present in raw data too subtle for traditional models to detect and usually discarded. However, the results are not ubiquitous.

Yan et al. pre-processed data by smoothing with a Savitsky-Golay filter and baseline removal via asymmetric weighted penalty least squares [37]. Compared to raw data, pre-processing improved the CNN accuracy to 98.75% from 96.70%.

Wu et al. found similar results. They performed baseline correction and spectral smoothing using the Vancouver Raman Algorithm and compared this regimen to raw data [35]. The processed regime outperformed the raw one across all subset analyses. This includes the traditional models, *k*-nearest neighbours (KNN), RF and SVM, but the largest increase in performance was observed in the CNN (accuracy: pre-processed = 81.3% vs. raw = 75.0%).

Although there is some suggestion that using CNNs could preclude an explicit pre-processing stage, this is not clearly established in the literature. There are a plethora of pre-processing techniques and it may be that some techniques were better suited to particular datasets than others—i.e., the pre-processing step could itself introduce over-fitting and CNNs trained on raw data would generalise better, even though their performance would be worse during training and testing. This is just speculation until it is more thoroughly explored. In all the above cases, where traditional ML models were used, they were improved by pre-processing. If pre-processing is to be conducted, the choices made to select the best method should be part of the model building process, essentially becoming another model hyper-parameter. The best pre-processing method should then be selected based on a validation set, and only tested on the test set once the final pre-processing method has been selected.

### 4.4. Traditional versus Deep Machine Learning

Due to the heterogeneity of the reviewed studies, no attempt was made to statistically aggregate the performance of traditional and deep learning models between studies. However, many studies performed such a comparison themselves. Table 2 compares the best performing traditional against deep models. Some studies have multiple entries as they compared performances across several data subsets or using several pre-processing strategies. Where more than two models were compared, the best-performing traditional and deep models are reported.

*Prima facie*, deep models consistently outperform traditional models, sometimes improving accuracy only by a few percent, but often by a large amount. However, given the methodological limitations discussed above, together with the propensity of more complex models to over-fit to data, particularly small datasets, this observation needs to be taken with caution. It should also be noted that deep learning models have more hyper-parameters to exploit, which can make them easier to overfit during hyper-parameter selection.

### 4.5. Model Transparency and Interpretability

ML studies, particularly deep learning, are often criticised for lacking interpretability, providing results without alluding to the underlying theory [9,54]. In this literature review, it was found that many studies attempted to relate classification results to the underlying biochemistry. Most did this by comparing the average spectra of different classes, and relating differences at given wavenumbers to the underlying antecedent biochemistry. However, this assumes that an ML model bases its classification on those features detectable by eye when looking at average spectra, which may or may not be valid. A more sophisticated method is employed when Raman peaks from different classes are statistically compared, or when the biochemical make-up of the samples is estimated with techniques such as NNLS. Regardless, these techniques only provide a post hoc narrative that may or may not reflect which spectral features the model utilised to drive classification. In general, it is not obvious what features a CNN is exploiting to make classifications.

Traditional models, being more simple, are often more amenable to interpretation. Riva et al. identified 19 Raman shifts as biochemically pertinent to classification via GB, as the technique involves an interpretable feature selection step [30]. Zuvela et al. were similarly able to identify relevant features due to the GA component of their model [40]. These, and other, traditional ML models have the advantage over deep learning models, which are infamous for their opacity. However, there are techniques available to help untangle such classifications, some of which were applied in the literature. Shu et al. applied a saliency map, in which 100 correctly classified spectra were randomly sampled from each class to which Gaussian noise was added [34]. This perturbation allowed for an assessment of which wavenumbers the model paid attention to. Koya et al. used a technique called permutation importance to assess the importance of features [25]. This randomly shuffles the features of input spectra and measures the corresponding change in prediction score. A significant drop in prediction suggested that the feature was particularly important to the model during classification.

### 4.6. Recommendations

There currently exists no explicit standard for conducting and reporting clinical applications of machine learning. However, reviews in adjacent medical domains have highlighted similar problems. A review of the literature focusing on deep learning applications assessed against clinicians found several areas to improve [55]. First, there was a lack of randomised controlled trials. This is also true of the RS literature reviewed above. Such a trial would require RS-based ML to be deployed in a clinical setting, and thoroughly assess how well the technology can generalise across settings. Significant drops in ML model performances are well documented when a model is applied to data collected at another institute [56]. Additionally, a large clinical trial would provide a firm benchmark of performance against human performance. All the studies reviewed here were non-randomised and none were compared to human performance, but rather used human pathological assessment as the gold standard to determine labels used for learning. If the technology is to translate to clinic then it must establish at least non-inferiority to expert humans whilst saving resources. Only two studies explicitly used more than one pathologist to provide labels [34,40]. Given the inter and intra-variability of even expert pathologists it would be prudent to seek consensus pathology to produce labels. This would also be true if ML is to be compared to human performance.

In addition, studies should detail enough of their methodology to be reproducible, which has been found to be lacking in adjacent domains [55]. In the context of this review, this includes all RS experimental parameters, all pre-processing steps (including data augmentation), the ML model, architecture and hyper-parameters used, and the cross-validation strategy used. Given that ML is a data-driven process, datasets and code could also be made available.

If the full potential of deep learning for biomedical RS is to be exploited, then large, publicly available datasets need to be curated. Despite the complexity of curating medical databases due to ethical concerns, such databases do exist for the development of deep learning models, such as the Cancer Genome Atlas [57]. No such database exists to examine RS in an oncology context. In addition to the practical challenges of collating large medical databases, RS poses the additional challenge of system transferability. This is relevant when a model is trained from data obtained on one spectrometer, then applied to data taken from a different spectrometer. Even with the same model of spectrometer from the same manufacturer, there will be subtle calibration differences in both frequency and intensity and differing fluorescence levels from the optical components. This becomes problematic when between-instrument variation exceeds between class variation [58]. This would be greatly aided by having consistent standards for system transfer, as advocated by Guo et al. [59]. It is worth highlighting that none of the studies reviewed here used multiple instruments, which would be almost essential in for any reasonable-scale practical application of the technique. This single fact underlines that, despite the many positive demonstrations, there are significant hurdles to overcome before such an application is realised in practice.

Taking all this into consideration, together with the literature reviewed here, we make the following recommendations:When data are split into training, validation and test sets, the level of the split should be conducted at the highest level (usually the subject, but this may be coincident with the sample). The chosen level should be explicitly stated.If model hyper-parameters are explored, including pre-processing, these should be selected based on a validation set and once selected only then tested against the test.All pre-processing and data augmentation techniques should be described in sufficient detail to allow for replication. While there is doubt regarding the need for pre-processing with CNNs, it would be beneficial for studies to include results from both raw and pre-processed data.All the hyper-parameters used in the model should be stated and the hyper-parameter search strategy should be reported.Where possible, consensus pathology should be used to confirm class labels.Where possible, attempts should be made to relate the results to the underlying biochemistry by interrogating model outputs, in addition to the traditional methods of comparing average spectra per class, peak comparisons and obtaining difference spectra.Resampling validation strategies should be used, such as *k*-fold and/or repeated CV. As these methods provide many generalisation estimates, standard deviations of performance metrics should be reported.

## 5. Conclusions

This review has found several common methodological issues in the recent literature that may be leading to an over-estimation of the ability of ML models to classify Raman oncology samples. Chief amongst them is the perennial problem in medical research of small datasets. This exacerbates several other issues including sub-optimal validation and sampling strategies such as using single splits instead of CV and failing to use a validation set when selecting hyper-parameters, including pre-processing methods and their attendant parameters. Regardless of the sample size, data are split into training/validation/test sets, but the highest hierarchical level in the data results in over-optimistic performance estimates. Thus, it is likely that the generally high performances seen in the literature are over-estimating true model performance. This is consistent with other medical domains employing ML [11].

This review is limited by searching English-language publications and by not by considering the grey literature, in particular conferences which provide rich sources of discussion. The review has also omitted several enhanced RS techniques such as SERS, sacrificing depth for brevity. Due to the heterogeneity of the studies, it was not possible to conduct a meta-analysis to provide a quantitative exploration of the literature. Contrary to the PRISMA guidelines, publication bias was not assessed. This is particularly difficult to assess in non-randomised trials and when meta-analysis has not been performed.

Future research could explore whether deep learning with sufficient sample sizes could make some of the numerous pre-processing techniques obsolete, or whether hand-crafted feature selection is an important junction in which to insert domain knowledge. Additionally, the use of data augmentation when using deep learning is relatively unexplored in the context RS and studies specifically looking into the facet would benefit the field. Regardless, most recommendations concern methodological issues. The reviewed papers are generally at the proof-of-concept stage, justifying their small sample sizes, but this necessitates more methodological rigour in order to draw accurate inferences. If the technique is ever to progress into accepted clinical practice, much larger studies need to be conducted, preferably populating a publicly accessible dataset, with the transferability between settings being established.

## Figures and Tables

**Figure 1 diagnostics-12-01491-f001:**
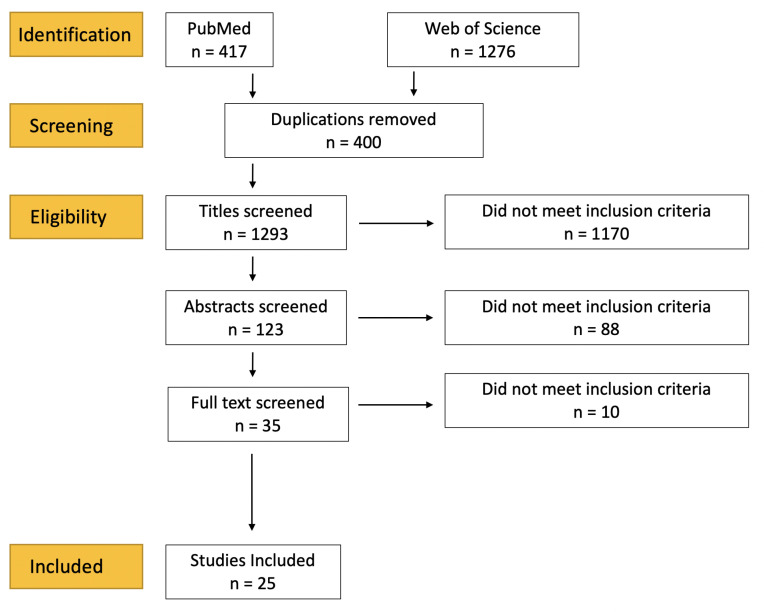
Literature search strategy: PRISMA flowchart of the literature selection process. *n* = number of studies.

**Figure 2 diagnostics-12-01491-f002:**
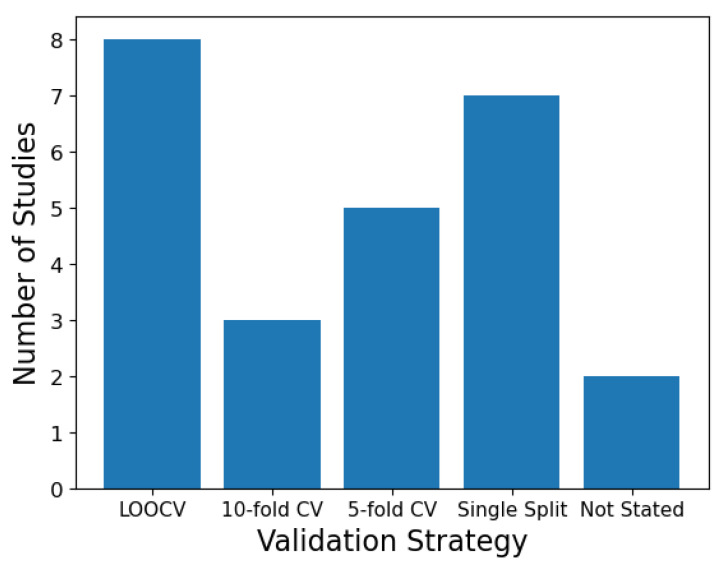
Validation strategy used in the reviewed literature. Some studies used more than one strategy.

**Figure 3 diagnostics-12-01491-f003:**
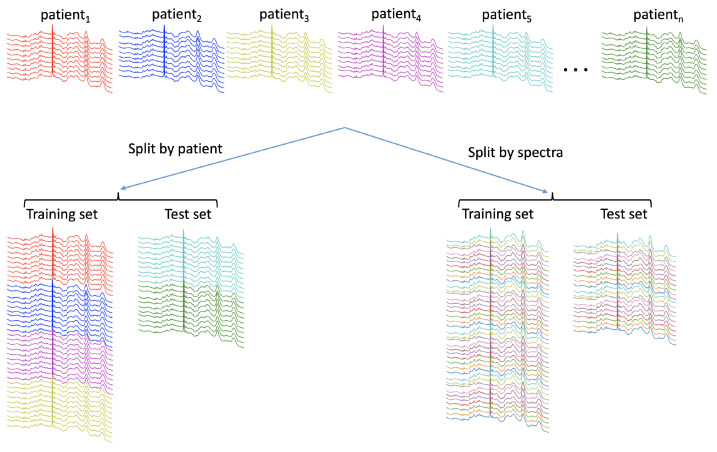
Spectra versus patient data splitting: note how the test set when split by spectra includes some spectra from all the patients contained in the train set.

**Table 1 diagnostics-12-01491-t001:** Literature review results: PCA—Principle Component Analysis, LDA—Linear Discriminant Analysis, QDA—Quadratic Discriminant Analysis, PLS—Partial Least Squares, SVM—Support Vector Machine, ANN—Artificial Neural Network, CNN—Convolutional Neural Network, RF—Random Forest, GB—Gradient Boost, CV—Cross Validation, LOOCV—Leave One Out Cross Validation, GA—Genetic Algorithm, NPC—Nasopharyngeal Carcinoma.

Authors/Year	Pathology Sample Type	Model	Validation Strategy	Number of Subjects/ Samples	Number of Spectra	Level of Split	Number of Classes	Accuracy (Sensitivity/ Specificity)
Aubertin et al., 2018 [16]	Prostate Cancer (tissue)	ANN	LOOCV	32 subjects/ samples	928	Not Stated	2	86% (87%/86%)
Baria et al., 2020 [17]	Skin Cancer (cell lines)	PCA-ANN	5-fold CV	Not Stated	150	Not Stated	3	96.7%
Bury et al., 2019 [18]	Brain Metastases (tissue)	PCA-LDA	Not Stated	21 subjects	525	Not Stated	2	80.2%
Chen et al., 2022 [19]	Ovarian Cancer (plasma)	ANN ensemble	Outer fold–single 66/33 Inner fold–5-fold CV	174 subjects	870	Spectra	2	94.8% (95%/95%)
Chen et al., 2021 [20]	Lung cancer & glioma (tissue)	CNN	5-fold CV	104 subjects/ samples	520 (2700 post augmentation)	Subject	2	99% (all pairwise comparisons > 95%
Daniel et al., 2019 [21]	Cervical Cancer (tissue)	PCA-ANN	Single 70/30	245 samples	Not Stated	Not Stated	3	99.0% (87%/86%)
He et al., 2021 [22]	Renal Cancer	SVM	LOOCV	77 subjects/ samples	4860	Subject	3	92.89%
Ito et al., 2020 [23]	Colon Cancer (serum)	Boosted Tree	Not Stated	184 subjects/ samples	3 spectra per subject. Average used	N/A	2	100%
Jeng et al., 2019 [24]	Oral Cancer (tissue)	PCA-QDA	*k*-fold CV and LOOCV	80 subjects/ samples	400	Sample	2	82% (84%/75%)
Koya et al., 2020 [25]	Breast Cancer (tissue)	CNN	Single split 60/20/20	88 subjects/ samples	34,505	Spectra	2	90% (89%—precision, 89%—recall)
Lee et al., 2020 [26]	Prostate Cancer (cell lines)	CNN	Single split 70/15/15	1 sample per class, 4 classes	300 (1200 post augmentation)	Spectra	4	97%
Ma et al., 2021 [27]	Breast Cancer (tissue)	CNN	10-fold CV	20 subjects, 40 samples	600 (5000 post augmentation)	Not Stated	2	92% (98%/86%)
Mehta et al., 2018 [28]	Brain Meningioma (serum)	PCA-LDA	LOOCV + independent test set	20 subjects, 70 samples	~8 spectra per subject. Average used	N/A	2	86%
Qi et al., 2022 [29]	Lung Cancer (tissue)	CNN	10-fold CV	77 subjects/ samples	15 spectra per sample	Spectra	2	98% (97%/99%)
Riva et al., 2021 [30]	Glioma (tissue)	GB	LOOCV	63 subjects/ samples	3450	Subject	2	83% (82%—precision, 82%—recall)
Santos et al., 2018 [31]	Skin (tissue)	PCA-LDA	Single split 60/40	128 samples	9–19 spectra per sample	Sample	2	62.5%
Sciortino et al., 2021 [32]	Glioma (tissue)	SVM	LOOCV	38 subjects/ samples	2073	Subject	2	87%
Serzhantov et al., 2020 [33]	Skin (tissue)	Gradient with soft voting	Single split 50/50, 1000 repeats	139 subjects	556	Not Stated	2	91% (93%/88%)
Shu et al., 2021 [34]	Nasopharyngeal Cancer (in vivo tissue)	CNN	10-fold Venetian Blind CV	418 subjects, 888 samples	15,354 (Augmented, quantity not specified)	Sample	2	84% (99%/66%)
Wu et al., 2021 [35]	Colon Cancer (tissue)	CNN	LOOCV	45 subjects/ samples	233 (2420 post augmentation)	Spectra AND Subject	3	94%—by spectra, 81%—by subject
Xia et al., 2021 [36]	Tongue Cancer (tissue)	CNN-SVM	5-fold CV	12 subjects, 24 samples	At least 216	Not Stated	2	99.5% (100%/100%)
Yan et al., 2021 [37]	Tongue Cancer (tissue)	CNN ensemble	5-fold CV	22 subjects, 44 samples	2004	Not Stated	2	99% (99%/98%)
Yu et al., 2021 [38]	Tongue Cancer (tissue)	CNN	5-fold CV	12 subjects, 24 samples	1440	Not Stated	2	97% (99%/94%)
Zhang et al., 2021 [39]	Breast Cancer (cell lines)	PCA-SVM	Single split	6 cell line 900 cells	4500	Not Stated	2	99.0% (100%/96%)
Zuvela et al., 2019 [40]	Nasopharyngeal Cancer (in vivo tissue)	GA- PLS-LDA	LOOCV	62 subjects, 113 samples	2126	Sample	2	98% (93%/100%)

**Table 2 diagnostics-12-01491-t002:** Deep versus traditional learning models. If the models were tested against various sub-sets of the data, this is given in the data subset column. Boldface text indicates the best-performing model.

Study	Deep Model	Traditional Model	Data Subsets
**Baria et al., 2020** [17]	96.0% (PCA-ANN)	**98.0%** (PCA-LDA)	SK-MEL-2 (Cell lines)
	**96.0%**	90.0%	SK-MEL-28
	**98.0%**	90.0%	MW-266-4
	**96.7%**	92.7%	All
**Daniel et al., 2019** [21]	**99.0%** (PCA-ANN)	98.0% (PCA-LDA)	
**He et al., 2021 [22]**	92.3% (ANN)	**92.9%** (SVM)	
**Lee et al., 2020 [26]**	**90.9%** (CNN)	78.3% (PCA-QDA)	Processed, whole spectra
	90.2%	**95.0%**	Processed, fingerprint
	**91.2%**	86.7%	Processed, high wavenumber
	**95.2%**	68.3%	Unprocessed, whole spectra
	**96.6%**	61.7%	Unprocessed, fingerprint
	**93.1%**	60.0%	Unprocessed, high wavenumber
**Ma et al., 2021 [27]**	**92.0%** (ANN)	86.5% (SVM)	
**Qi et al., 2022 [29]**	**97.7%** (ANN)	86.6% (PCA-LDA)	Adenomcarcinoma
	**96.1%**	82.1%	Squamous cell carcinoma
**Shu et al., 2021 [34]**	**82.1%** (CNN)	73.6% (PLS-LDA)	All data
	**84.4%**	83.7%	NPC vs. control
	**82.1%**	68.4%	NPC vs. post-treatment
**Wu et al., 2021 [35]**	**81.3%** (CNN)	52.7% (KNN)	Processed data
	**75.0%**	42.0% (SVM)	Unprocessed data
**Xia et al., 2021 [36]**	**99.6%** (CNN-SVM)	95.4% (PCA-SVM)	
**Yan et al., 2021 [37]**	**98.8%** (Ensemble CNN)	88.5% (PCA-SVM)	
**Yu et al., 2021 [38]**	**96.9%** (CNN)	88.5% (SVM)	

## Data Availability

No new data were created or analysed in this study. Data sharing is not applicable to this article.

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
