# Peer review of "Machine Learning of Raman Spectroscopy Data for Classifying Cancers: A Review of the Recent Literature"

_diagnostics, 2022, doi:10.3390/diagnostics12061491_

Round 1

Reviewer 1 Report

Review of “Machine Learning of Raman Spectroscopy Data for Classifying Cancers: a Review of the Recent Literature” by Nathan Blake, Riana Gaifulina, Lewis D. Griffin, Ian M. Bell and Geraint M. H. Thomas

The authors present a review of recent peer reviewed literature to ascertain the performance of both traditional machine learning methods and deep learning methods recently applied to Raman spectra of cancers. They present their finding of a number of methodological considerations which may lead to an over-estimation of the performance of deep learning models; primary among these was small sample sizes, which compound upon sub-optimal choices regarding sampling and validation strategies. They recommend the collation of large benchmark Raman datasets in order to increase sample sizes that are accessible for researchers to develop and refine deep learning models.

The review is well written and clearly presented, and includes all the basic data pre-processing steps required for Raman spectra, such as smoothing, derivative/slope recognizers, thresholding and spectral region selection to address variation in background signals and noise and variable selection. It also covers different validation strategies. The figure and tables are well presented, however, the font in Table 1 font needs to be enlarged.

The paper includes a wide variety of cancers and provides a review of the literature on machine learning methods applied to Raman spectra of these between 2018 – 2022.  However, it is not clear what is meant by “traditional machine learning methods” and whether the aim of the study is to compare traditional machine learning methods (I assume these include ANN),  classical linear methods (e.g. PCA-DA, PLS-DA, SIMCA?) and deep learning methods (e.g., CNN?). Have all classical methods in this period been included in the study? Reference is made to traditional machine learning methods, classical linear methods and deep machine learning methods, and in some sections linear models are differentiated from ML models. This requires clarification, as indicated in the comments below. There are also a number of other concerns that need to be addressed, also given in the comments below.

Once these have been satisfactorily addressed, the paper will be suitable for publication.

1)     Abstract: other factors inhibiting the use of RS in the clinical setting need to be commented on besides the complexity of the data, such as cost and lowered signal to noise ratio for portable systems 

2)     Please define what is meant by machine learning methods. Throughout the paper, “traditional machine learning methods” are mentioned. It is not clear whether the aim of the study is to compare traditional machine learning methods (e.g. ANN),  classical linear methods (e.g. PCA-DA) and deep learning methods (e.g., CNN)? Table 2 suggests the traditional machine learning methods are classical linear methods. The different terms should be defined in the introduction, with examples of the different methods given. This is particularly relevant to the comment number 5 below, where linear models are differentiated from ML models.

3)     Page 2, line 76 must be corrected: reference 12 in fact showed that the resulting minimum CV error estimate is not an unbiased estimate of the true error that can be expected from the final classifier on independent data. They instead concluded that the use of  a nested CV procedure (in which an inner CV loop is used to perform the tuning of the parameters with an outer CV being used to compute an estimate of the error) is needed to provide a more unbiased estimate of the true error.

4)     Page 8, line 334: define what is meant by “single splits”: does this include the “Venetian blind” method with k-fold blinds? Related to this, a discussion of random k-fold CV vs the Venetian blind CV and LOOCV should be included.  

5)     Page 10 lines 383: please elaborate on how ML can account for heterogeneity when the individual spectra are used as opposed to the mean, in the sentence “It is arguable that this is also taken into account when all spectra are used in a ML model”. Give examples of the “linear models” referred to in line 385

6)     Page 10, Line 397: again, please elaborate on the sentence in lines 396 – 397, on why this does not extend to ML?

7)     Page 10, section 4.1.2 line 392:  this discussion is too general. The need for paired sampling very much depends on the type of cancer and the biomarkers associated with it, and the conclusion cannot be generalized. If a cancer can clearly be defined by the presence or change in a single biomarker then paired sampling may not aid ML. However, if changes to different extents in number of different biomarkers and other factors associated with an individual are relevant (for e.g., ethnicity, sex, age, genetics) then this could not be the case. Also, the relevant biomarkers are not always identifiable by some ML.

8)     Page 12, section 4.2. The effect of data augmentation of Raman spectra of biological samples on classification accuracy needs further investigation. Adding artificial noise, baseline offsets and slopes does not add to the variation needed in the biomarker dataset, effectively only adding “more of the same” biomarker information. The inherent variation in biological samples requires sufficient numbers of samples to ensure the dataset incorporates enough of the variation that will be encountered in the application in the clinic.  This should be commented on.

9)     Increase the font size of Table 1

10)  Page 15, section 4.6, lines 612-620. The difficulties needed to be addressed and overcome in setting up such a database include the challenges of instrument differences (for e.g. spectrometer components in different makes of spectrometers which affect spectral resolution, detector type and sensitivity, excitation wavelength used (resulting in varying relative peak intensities and peaks present and levels of fluorescence) and calibration). These need to be discussed.

Referencing errors and minor typos

1.     References: 1 and 2 : add the other authors referred to by “et al.”

2.     Page 12 line 490: reference 49 is incorrect – should be 54

3.     Page 7 line 244: “lent” should be “leant”

4.     Page 15, line 623: “train” should be “training”

Author Response

Thank you for your detailed comments, which have helped us improve the manuscript. We have addressed all your numbered points below starting with repeating your comments, our quick reply, which page/line in the revised manuscript the changes occur and in italics the changes themselves.

1.) Abstract: other factors inhibiting the use of RS in the clinical setting need to be commented on besides the complexity of the data, such as cost and lowered signal to noise ratio for portable systems 

This has been added to the introduction:

Page 2 Line 56:

Aside from modelling issues, a number of practical issues remain, such as establishing cost-effectiveness, the choice of substrate on which to mount \emph{ex vivo} samples and the miniaturisation of the technology for \emph{in vivo} testing while maintaining signal-to-noise \cite{baker2018clinical}. Though important for RS to become established in clinical practice, these will not be discussed in this review.

2.) Please define what is meant by machine learning methods. Throughout the paper, “traditional machine learning methods” are mentioned. It is not clear whether the aim of the study is to compare traditional machine learning methods (e.g. ANN),  classical linear methods (e.g. PCA-DA) and deep learning methods (e.g., CNN)? Table 2 suggests the traditional machine learning methods are classical linear methods. The different terms should be defined in the introduction, with examples of the different methods given. This is particularly relevant to the comment number 5 below, where linear models are differentiated from ML models.

A more precise definition of traditional and deep learning has been added in the introduction based on the universal approximation theorem. By this definition, many traditional ML models happen to be linear, but it is not a defining feature.

Page 2 Line 44:

In particular, we define a deep learning model as any model based on neural network architecture, from ANNs to more sophisticated structures such as CNNs. Traditional ML models include all other models, whether they are linear, such as LDA, or non-linear, such as SVMs with an appropriate kernel function.  

3.)  Page 2, line 76 must be corrected: reference 12 in fact showed that the resulting minimum CV error estimate is not an unbiased estimate of the true error that can be expected from the final classifier on independent data. They instead concluded that the use of a nested CV procedure (in which an inner CV loop is used to perform the tuning of the parameters with an outer CV being used to compute an estimate of the error) is needed to provide a more unbiased estimate of the true error.

We agree with your assessment and believe we were trying to convey the same thought. The wording has been changed to be more clear and a more precise reference given:

Page 2 Line 79:

The validation set is used to optimise model hyper-parameters (or even guide the choice of ML model): choices about the model that a researcher makes which influence its classification ability. Similar to how a model can over-fit the data, the hyper-parameters can be selected such that it performs well for a given test set, but fails to generalise well. This has been described as over-fitting at the second level of inference cite{cawley2010over}. The validation set provides an additional set to allow the hyper-parameter space to be optimised, while preserving the test set for a less biased estimate of the generalisability.

4.)  Page 8, line 334: define what is meant by “single splits”: does this include the “Venetian blind” method with k-fold blinds? Related to this, a discussion of random k-fold CV vs the Venetian blind CV and LOOCV should be included.  

A sentence distinguishing between single split and CV has been added:

Page 11 Line 347:

Single split validation refers to the method of dividing the data into different sets just once, as opposed to CV methods which iteratively split the data to give several performance estimates. 

A discussion around Venetian Blinds CV has been added:

Page 10 Line 336:

Standard $k$-fold CV splits data to folds at random. Shu emph{et al.} employed a variant called Venetian Blinds, which systematically assigns data to folds cite{BigTaiwanNaso}.There is some suggestion that the method performs well compared to random, and other, methods, though this depends upon the ML model used cite{racz2018modelling}, and very likely the dataset in question. Overall it has received little attention in the literature, and its utility to Raman datasets unexplored.

5.) Page 10 lines 383: please elaborate on how ML can account for heterogeneity when the individual spectra are used as opposed to the mean, in the sentence “It is arguable that this is also taken into account when all spectra are used in a ML model”. Give examples of the “linear models” referred to in line 385

The paragraph has been changed to accommodate your points, with the claim about linear models being withdrawn as there is no clear evidence that is the case:

Page 12 Line 397:

There are two methods by which sample heterogeneity is currently incorporated in the literature. Averaging spectra provides one means, while including multiple spectra from a single sample provides another. There has been no direct comparison of these methods in the reviewed literature.  A priori there is no reason to suspect one will work better than the other and will likely depend upon the intended application. For instance, a mean spectrum is typically of higher signal-to-noise than individual spectra, and a model trained on the former may not generalise to applications requiring individual spectra to be classified, such as post-surgery cancer edge detection. It is also an open question whether averaging spectra from a sample before analysis would be an effective use of data for deep learning models, which are notoriously data intensive. Overall, it is not clear that averaging spectra will always provide a benefit, and the decision to do so should take into account the nature of the application and the ML model being used. 

6 -7.)

Page 10, Line 397: again, please elaborate on the sentence in lines 396 – 397, on why this does not extend to ML?

Page 10, section 4.1.2 line 392:  this discussion is too general. The need for paired sampling very much depends on the type of cancer and the biomarkers associated with it, and the conclusion cannot be generalized. If a cancer can clearly be defined by the presence or change in a single biomarker then paired sampling may not aid ML. However, if changes to different extents in number of different biomarkers and other factors associated with an individual are relevant (for e.g., ethnicity, sex, age, genetics) then this could not be the case. Also, the relevant biomarkers are not always identifiable by some ML.

These have been addressed together.

Clarification on why paired testing does not automatically extend to ML provided. 

We have also expanded the discussion to include some potential pitfalls and benefits of paired sampling, which hopefully address your point.

Page 12 Line 414:

This may be true of traditional statistical set-ups, such as hypothesis testing, but does not necessarily extend to ML. In hypothesis testing, paired sampling allows a single mean and variance to be calculated: that of the difference between paired samples. A similar procedure is not typically conducted in ML, including the literature reviewed here. Rather, even if paired samples have been taken, they are treated as independent samples (analogous to paired samples being taken but a normal, non-paired, statistical hypothesis test being performed). Additionally, by using paired data it could be argued that the training sample is not as inclusive of the general population, denying the model the opportunity to distinguish ever more subtle differences between cancerous vs healthy tissues. Again, it depends upon the clinical application - will classification be made on the basis of comparing a patient's healthy sample to their own suspected diseased sample? If not then a paired training regime does not reflect the clinical application, and will compromise generalisability. However, ML models could be adjusted to account for paired samples, and may be particularly useful for longitudinal studies and disease progression tracking, where the task is explicitly to compare a single patient's sample against their past samples. 

8.)  Page 12, section 4.2. The effect of data augmentation of Raman spectra of biological samples on classification accuracy needs further investigation. Adding artificial noise, baseline offsets and slopes does not add to the variation needed in the biomarker dataset, effectively only adding “more of the same” biomarker information. The inherent variation in biological samples requires sufficient numbers of samples to ensure the dataset incorporates enough of the variation that will be encountered in the application in the clinic.  This should be commented on.

The discussion has been changed to clarify that data augmentation is an algorithmic concern (i.e. to prevent over-fitting), and cannot overcome a lack of biological variation in the original data as you point out:

Page 14 Line 490:

This process does not seek to simulate biological variation, but is useful as a regularisation technique, which lessens the tendency of a model to over-fit the data. Hence, augmentation cannot be used to compensate for a lack of biomedical diversity in the data. 

9.) Increase the font size of Table 1

Adjusted

10.) Page 15, section 4.6, lines 612-620. The difficulties needed to be addressed and overcome in setting up such a database include the challenges of instrument differences (for e.g. spectrometer components in different makes of spectrometers which affect spectral resolution, detector type and sensitivity, excitation wavelength used (resulting in varying relative peak intensities and peaks present and levels of fluorescence) and calibration). These need to be discussed.

This crucial point has now been acknowledged:

Page 17 Line 636:

In addition to the practical challenges of collating large medical databases, RS poses the additional challenge of system transferability. This is relevant when a model is trained from data obtained on one spectrometer, then applied to data taken on a different spectrometer. Even with the same model of spectrometer from the same manufacturer, there will be subtle calibration differences in both frequency and intensity and differing fluorescence levels from the optical components. This becomes problematic when between instrument variation exceeds between class variation \cite{guo2017towards}. This would be greatly aided by having consistent standards for system transfer, as advocated by Guo \emph{et al.} \cite{guo2021chemometric}. It is worth highlighting that none of the studies reviewed here used multiple instruments, which would be almost essential in for any reasonable-scale practical application of the technique. This single fact underlines that, despite the many positive demonstrations, there are significant hurdles to overcome before such an application is realised in practice.

Reviewer 2 Report

Paper deals with important task. The authors conducted a systematic literature review of applying the ML-based methods for solving the task of cancer classification using Raman Spectroscopy data.

Paper has great practical value in Medicine.

It has a logical structure, all necessary sections. The authors used PRISMA scheme using PubMed and WoS databases. This is the advantage of this paper. Paper is technically sound. The analysis concerns various aspects of the application of machine learning algorithms to solve the problem. This is the second big advantage of this paper.

Suggestions:

  1. Introduction section should be extended using non-iterative ANN's to solve the stated task. The authors can use SGTM neural-like structure and its modifications.
  2. It would be good to add point-by-point the main findings in the end of the Introduction section
  3. It would be good to add the remainder of this paper
  4. Table 1 is too small. Please improve it.
  5. The Conclusion section should be extended using: 1) main findings obtained in the paper; 2) limitations of the conducted review; 3) prospects for the future research.

Author Response

Thank you for the feedback. We have have addressed your points below by repeating your comments, our response, the page/line of any changes in the new manuscript and in italics the changes themselves.

1.)  Introduction section should be extended using non-iterative ANN's to solve the stated task. The authors can use SGTM neural-like structure and its modifications.

If by non-iterative NNs you mean RNNs, these have been mentioned. SGTMs were not used in the reviewed literature, so we didn’t consider it relevant to discuss them specifically.

2.)  It would be good to add point-by-point the main findings in the end of the Introduction section

We have improved table 1 and added extended the conclusion section as per points 4 and 5. We hope this extra clarification suffices as we do not want the paper to become too repetitive.

3.)  It would be good to add the remainder of this paper

We didn’t really understand this comment, but hope that our response to your other points and other general improvements will satisfy this concern.

4.)  Table 1 is too small. Please improve it.

Improved

5.)  The Conclusion section should be extended using: 1) main findings obtained in the paper; 2) limitations of the conducted review; 3) prospects for the future research.

The Conclusion section has been extended to provide more clarity and include the suggested sub-sections. 

Page 18 Line 671:

This review has found several common methodological issues in the recent literature that may be leading to an over-estimation of the ability of ML models to classify Raman oncology samples. Chief amongst them is the perennial problem in medical research of small datasets. This exacerbates several other issues including sub-optimal validation and sampling strategies such as using single splits instead of CV and failing to use a validation set when selecting hyper-parameters, including pre-processing methods and their attendant parameters. Regardless of the sample size, data split into training/validation/test sets at anything but the highest hierarchical level in the data results in over-optimistic performance estimates. Thus it is likely that the generally high performances seen in the literature are over-estimating true model performance. This is consistent with other medical domains employing ML \cite{covid}.

This review is limited by searching English language publications and by not by considering the grey literature, in particular conferences which provide rich sources of discussion. The review has also omitted several enhanced RS techniques such as SERS, sacrificing depth for brevity. Due to the heterogeneity of the studies it was not possible to conduct a meta-analysis to provide a quantitative exploration of the literature. Contrary to the PRISMA guidelines, publication bias was not assessed. This is particularly difficult to assess in non-randomised trials and when meta-analysis has not been performed.

Future research could explore whether deep learning with sufficient sample sizes could make some of the numerous pre-processing techniques obsolete, or whether hand-crafted feature selection is an important junction in which to insert domain knowledge. Also, the use of data augmentation when using deep learning is relatively unexplored in the context RS and studies specifically looking into the facet would benefit the field. Regardless, most recommendations concern methodological issues. The reviewed papers are generally of the proof of concept stage, justifying their small sample sizes, but this necessitates more methodological rigour in order to draw accurate inferences. If the technique is ever to progress into accepted clinical practice much larger studies need to be conducted, preferably populating a publicly accessible dataset, with the transferability between settings being established.

Reviewer 3 Report

Thanks for your review on the classification of cancer. this is a well-documented review however some minor edits are needed before publishing:

(1) Keywords need to be more generic and need to be increased.

(2) Figure 1: caption needed to be more explanatory.

(3) Table 1: It is unacceptable in its current form. Please either redraw it in a more compact way or split it into multiple ones if not possible in one. Currently, it is not visible when printed.

(4) For each type, there needs to be a table at the end of the section and text must be reduced in order to save the reader time.

(5) Table 2 is perfect and self-explanatory.

Author Response

Thank you for the feedback. We have addressed you points below by repeating your comments, our response, the page/line of any changes and in italics the changes themselves.

(1) Keywords need to be more generic and need to be increased.

Keywords increased:

Page 1 Line 14:

Raman Spectroscopy; Medical application; Disease screening and diagnosis; Machine learning, Cross-Validation, Deep learning

(2) Figure 1: caption needed to be more explanatory.

Additional information provided in caption:

Page 3 Line 105:

Literature Search Strategy: PRISMA flowchart of the literature selection process. $n$ = number of studies.

(3) Table 1: It is unacceptable in its current form. Please either redraw it in a more compact way or split it into multiple ones if not possible in one. Currently, it is not visible when printed.

Redrawn.

(4) For each type, there needs to be a table at the end of the section and text must be reduced in order to save the reader time.

We have improved table 1 as suggested and hope that is sufficient to address your concerns. We did not want to add an additional table for each section to save on repetition, and as most literature reviews place all this information in a single table.

Round 2

Reviewer 1 Report

The authors have addressed my queries and updated the paper adequately. The paper is suitable for publication in its revised form.